# Ramadan during pregnancy and neonatal health—Fasting, dietary composition and sleep patterns

Fabienne Pradella[1]*, Birgit Leimer[1], Anja Fruth[2], Annette Queißer-Wahrendorf[3], Reyn Joris van Ewijk[1]

1 Chair of Statistics and Econometrics, Faculty of Law, Management and Economics, Johannes Gutenberg-University, Mainz, Germany , 2 Department of Obstetrics and Gynecology, University Medical Center of Johannes Gutenberg-University, Mainz, Germany, 3 Center for Pediatric and Youth Medicine, University Medical Center of Johannes Gutenberg-University, Mainz, Germany

☯ These authors contributed equally to this work.
* fabienne.pradella@uni-mainz.de

## Abstract

### Background and objectives

Large shares of pregnant Muslims worldwide observe the Ramadan fast. Previous research showed that Ramadan during pregnancy is associated with adverse offspring health outcomes lasting throughout the life-course. Evidence on effects on birth outcomes is inconclusive, however, and previous research did not consider the role of dietary composition and sleep patterns during Ramadan. This study systematically documents maternal lifestyle during Ramadan and assesses if diet and sleep adaptations to Ramadan, independent of and in addition to maternal fasting, are associated with neonatal health outcomes.

### Methods

This study reports a survey of 326 Muslims who delivered their baby in Mainz, Germany, linked to maternal & infant hospital records. Participants reported on fasting, dietary composition and sleep schedules while pregnant during Ramadan.

### Results

Fasting during pregnancy was associated with reduced birthweight, in particular for fasting during the first trimester (-352·92g, 95% CI: -537·38; -168·46). Neither dietary composition nor altered sleep were directly associated with birthweight. However, dietary composition during Ramadan outside of fasting hours seems to moderate the fasting-birthweight association, which disappeared for women switching to high-fat diets.

### Conclusions

The finding that dietary intake during Ramadan potentially moderates the fasting-birthweight association is of high relevance to pregnant Muslims who wish to fast and their healthcare professionals, since dietary choices outside of fasting hours are often relatively easily

**Data Availability Statement:** This study is based on a relatively small dataset and our data includes information on study participants that, in combination, may be identifying. In particular, to

calculate variables that are crucial to our analysis (such as overlap with Ramadan during pregnancy and maternal age), sensitive information including the birth dates of both mother and child are necessary. Moreover, since only a limited number of children are born each day in Mainz, and Muslims often can be identified by their surname, making the data publicly available would risk participant privacy. Access to the data is possible for researchers who meet the criteria for access to confidential data via the ethics committee of Johannes Gutenberg-University Mainz, Gutenberg School of Management and Economics (https://en.wiwi.uni-mainz.de/ethics-committee/).

**Funding:** This research was funded by the German Research Foundation (DFG, https://www.dfg.de/), grant 260639091 (awarded to RvE). The funders had no role in study design, data collection and analysis, decision to publish, or preparation of the manuscript.

**Competing interests:** The authors have declared that no competing interests exist.

**Abbreviations:** BMI, Body Mass Index.

modifiable. This is the first study to include information on maternal diet and sleep during Ramadan, and additional research is needed to assess the roles of specific (macro)nutrients and food groups.

## Introduction

Intermittent fasting is a widespread practice among women in childbearing age, including pregnant women [1–3]. One form of intermittent fasting during pregnancy is adherence to the Ramadan fast. During Ramadan, which lasts for 29–30 days, adult Muslims abstain from food and drinks during daylight hours. Most Muslim pregnancies overlap with a Ramadan and many pregnant Muslims decide to fast. Estimated rates of fasting among pregnant Muslims range from 54% in the Netherlands to 87% in Pakistan and Singapore and 99% in Bangladesh [4–7]. While the literature on Ramadan fasting during pregnancy and birth outcomes has remained inconclusive [8], various studies demonstrated that being born in the months after a Ramadan is predictive of worse later-life cognitive and physical health outcomes among Muslims, including symptoms of pulmonary disease, coronary heart disease and type 2 diabetes, increased disability rates, and worse performance in school and on the job market [9–16]. It is generally assumed that maternal intermittent fasting is the driver of the associations with offspring health. However, adherence to the Ramadan fast entails further lifestyle changes to dietary and sleep patterns, which might also impact offspring health. These adjustments to Ramadan can occur independent of the fasting decision, as non-fasting pregnant Muslims often live in households with fasting members [17, 18].

While diets during non-fasting hours and sleeping patterns have been hypothesized as further channels or moderators for the health effects of Ramadan during pregnancy, the previous literature did not have the necessary data to investigate [11, 18, 19]. Dietary intake during non-fasting hours is characterized by traditional meals that often have higher contents of fat and simple sugars than meals outside of Ramadan, in particular during the breaking of the fast after sunset. Sleep schedules are adjusted during Ramadan since dietary intake and food preparation are shifted to night hours. This study provides first evidence on how Ramadan-related lifestyle beyond maternal fasting is associated with birth outcomes. We compare birth outcomes of Muslim women who did versus did not fast, adapt their dietary intake (simple sugars, fat and fluids) and adjust their sleep rhythm during Ramadan. We use detailed survey data on 326 Muslim women whose pregnancies overlapped with the same Ramadan and who delivered in Mainz (Germany), in combination with hospital-based information on their newborns' health at birth.

## Methods

### Recruitment of study participants

All Muslims who delivered their singleton newborn in one of the two obstetric wards in Mainz and whose pregnancy overlapped with Ramadan 2017 were eligible for participation. Muslim women without overlap with Ramadan 2017 were excluded from the sample, as well as women who preregistered for delivery but did not deliver their baby in Mainz (Fig 1). Multiple births were excluded from the regression analysis since multiple births form a special group in terms of prenatal care provision as well as neonatal health outcomes, including low birth weight and lower gestational age [20]. Due to the high participation rate among the relevant population

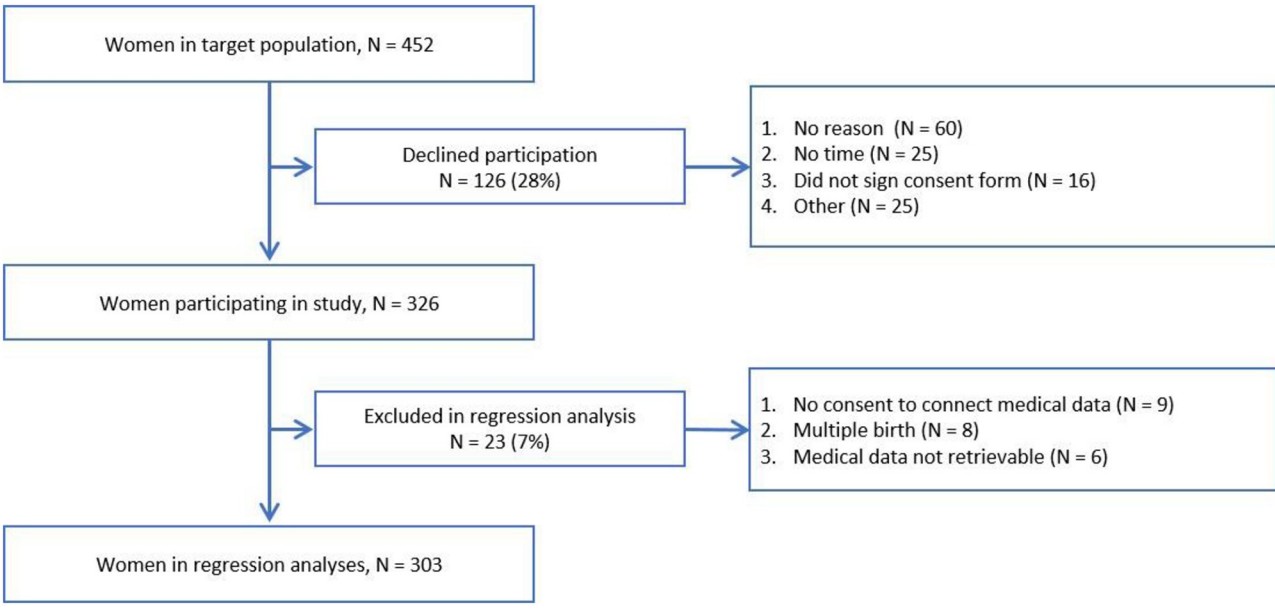

**Fig 1. Sample selection.** Flow diagram of the sample recruitment.

(72%), our sample is representative of pregnant Muslim women delivering in Mainz, speaking German, Arabic, Turkish or English.

The survey consisted of interviews using a structured questionnaire (S1 File) and was conducted in German, Arabic, Turkish and English, to collect information on fasting, dietary choices and sleeping habits during Ramadan and maternal background characteristics. Upon consent, survey data was linked to hospital medical information on neonatal health outcomes.

Before this study, a pilot study with 116 participants was carried out [17]. The procedures to approach the relevant population and the questionnaire were tested, revised and validated. The ethics committee of the State Chamber of Medicine in Rhineland-Palatinate (Germany) reviewed and approved this study (837.309.14 (9548)). All participants gave written informed consent before taking part in this study.

## Outcomes of interest

The primary health outcome studied is birthweight (in grams), which has been shown to be predictive of outcomes along the entire life span [21–23]. As an alternative variable, length of gestation is used. Gestational length is considered only a secondary variable as it is only available as fully completed weeks, leading to a loss of detail. Moreover, there is less evidence that variation within the normal range is predictive of short- and long-term health. Using a binary variable for prematurity would lead to a decreased statistical power.

## Exposure measures

Exposure to maternal Ramadan fasting is defined as being born to a mother who fasted at least 3 days during pregnancy in order to capture the associations between regular fasting and offspring health. In a robustness check, we define exposure to fasting as being born to a mother who fasted at least 1 day.

We furthermore differentiate exposure to maternal Ramadan fasting by pregnancy trimester of exposure. The trimester during which Ramadan occurred is determined based on

the estimated date of conception from hospital birth records. If exposure falls within two trimesters, the observation is assigned to the trimester where the number of days of overlap is larger. We also differentiate by intensity of fasting, categorizing mothers as fasting on some days (3–9 days), around half of Ramadan (10–19 days) versus on most or all days (20–29 days).

Sleep schedules are adapted during Ramadan as meals and their preparation are shifted to night hours. Previous research suggests that lack of maternal sleep or poor maternal sleep quality may be associated with poorer birth outcomes [24]. We define maternal sleep reduction as having slept less than in the month before Ramadan, i.e. a mother is considered to have slept less if she reported going to bed later or getting up earlier (or both) during Ramadan compared to the month before, while not napping more during the day.

With respect to maternal dietary adjustments to Ramadan, we assess the role of altered intakes of simple sugars, fat and fluids. Traditional meals during Ramadan generally tend to have high contents of simple sugars. Previous research documented that pregnant women skipping meals experience the phenomenon of accelerated starvation, describing an increased susceptibility to ketosis in response to low blood sugar levels [25, 26]. Due to the high glycemic index of foods rich in simple sugars, it appears likely that an increased intake of foods rich in simple sugars during Ramadan might increase the likelihood for accelerated starvation to occur. For these reasons when studying how Ramadan during pregnancy affects birth outcomes, it is on the one hand relevant to assess the role of an altered intake of foods with a high sugar itself, and on the other hand, to assess whether a high sugar intake food moderates the effect of fasting on birth outcomes. I.e. an increased intake of high glycemic index foods may exacerbate effects of fasting. We categorize mothers as having either consumed fewer, the same amount or more foods rich in simple sugars, as compared to the month prior to Ramadan.

If a woman decides not to fast, changes in her fat and fluid intake during Ramadan are unlikely to affect her birth outcomes. But among fasting women, they may moderate the associations between fasting and offspring health outcomes. We therefore include measures of high-fat content diets and fluid intake among fasting women in moderation analyses. In this analysis, we include information on whether fasting mothers reported having eaten more, less or the same amount of foods high in fat content compared to the month before Ramadan.

Research has shown reductions of body fluid compartments among fasting Muslims [27]. During pregnancy, the required water intake is increased and dehydration has been shown to lead to lower amniotic fluid levels, which have in turn been found to be associated with adverse birth outcomes [28, 29]. In order to assess if the fasting-birth outcomes associations vary by maternal fluid intake during non-fasting hours, we include information on whether fasting women reported drinking more, less or the same amount compared to the month before Ramadan in the moderation analysis.

## Covariates

The variables selected as covariates were sex of the offspring, gestational week at birth (week and week squared) and maternal age at birth (age, $age^2$ and $age^3$). We further control for maternal employment status prior to parental leave, highest educational attainment, country of birth, nulliparity, indicator for length of stay in Germany (fewer or more than 3 years), being more religious (measured as fasting during Ramadan when not pregnant and using veiling on a day-to-day basis), pre-pregnancy body mass index (BMI), pregnancy risk factors (smoking, alcohol consumption, drug use, consanguinity) and awareness of the pregnancy during Ramadan.

## Statistical analysis

First, linear regressions are estimated to identify the fasting-birth outcomes association, including all covariates. Second, we additionally adjust for maternal shorter sleep duration and intake of foods rich in simple sugars to investigate whether these are directly associated with birth outcomes, or whether the fasting-birth outcomes association may have been confounded by altered sleep and sweet foods consumption. Third, we include interaction terms between fasting and sleep and dietary intake patterns to analyze the latter's potential moderating role.

## Robustness and heterogeneity

In an additional analysis, we interact trimester-specific exposure with sleep and dietary intake patterns in order to investigate if potential associations are concentrated in specific pregnancy phases. To test the sensitivity of our results, we run regressions in which maternal fasting is defined as having fasted at least 1 day. We also run analyses in which the sample is reduced to full-term pregnancies (≥37 weeks of gestation) and to normal-term pregnancies (≥37 & ≤42 weeks of gestation). In order to account for the possibility of residual confounding, we calculate the Oster test statistic (see S2 File for methodological details).

## Results

### Descriptive statistics

The study population consists of the cohort of 326 Muslim women who delivered in Mainz (capital of the German state Rhineland-Palatinate) and whose pregnancy overlapped with the Ramadan in the study year.

Of all interviewees, 30% reported having fasted during pregnancy (Table 1). 47% of the fasting women reported fasting at least 20 days, while the fasting rate is highest in the first pregnancy trimester (Fig 2).

Women who fast during pregnancy are likely to be more religious, to have lived in Germany for less than three years, not to be employed, and not to have been aware of their pregnancies during Ramadan (Table 2). In raw mean comparisons, birthweight is slightly, but insignificantly, lower among fasting women.

### Associations between fasting and birth outcomes

Offspring to mothers who fasted had lower birthweights compared to offspring of non-fasting women (-158·19g, 95% CI: -300·83; -15·55) (Fig 3). In particular, children of mothers fasting during the first trimester had significantly lower birthweights than children of mothers who experienced a Ramadan during the first trimester but did not fast (-352·92g, 95% CI: -537·38; -168·46). Effects of fasting 10–19, and 20–29 days were similar in size, while the association between fasting 3–9 days and birthweight was considerably smaller in size and not significant.

Adding sleep reduction and sweet food consumption to the adjusted regression model did not alter the magnitude of the fasting-birthweight association (Fig 4). Neither sleep reduction nor sweet food consumption were themselves significantly associated with birthweight.

The same analyses were conducted with gestational age as the dependent variable. No significant associations were found (see S1 and S2 Figs).

### Moderating effects of sleep and dietary patterns

Fig 5 shows that the association between fasting and birthweight appears considerably moderated by maternal dietary intake during non-fasting hours. Particularly, the negative fasting–

**Table 1. Fasting, sleep & nutrition during Ramadan: Comparison of fasting and non-fasting women using univariate analysis.**

| Category | Total Sample (N = 326) | | Fasting Women (N = 98) | | Non-Fasting Women[1] (N = 207) | | p-Value for Diff. |
|---|---|---|---|---|---|---|---|
| | Obs. | Share | Obs. | Share | Obs. | Share | |
| **Fasting Behavior** | | | | | | | |
| Fasted | 98 | 30% | 98 | 100% | | N/A | |
| Fasted 3–9 days | 24 | 7% | 24 | 24% | | | |
| Fasted 10–19 days | 28 | 9% | 28 | 29% | | | |
| Fasted ≥ 20 days | 46 | 14% | 46 | 47% | | | |
| **Trimester of Ramadan Occurrence during Pregnancy** | | | | | | | |
| Trimester 1 | 117 | 36% | 49 | 50% | 60 | 29% | |
| Trimester 2 | 92 | 28% | 24 | 24% | 63 | 31% | 0.001 [2] |
| Trimester 3 | 116 | 36% | 25 | 26% | 83 | 40% | |
| **Sleep & Diet during Ramadan** | | | | | | | |
| Slept less | 124 | 39% | 42 | 43% | 78 | 38% | 0.442 [2] |
| Decreased sweet foods intake | 68 | 21% | 25 | 25% | 39 | 20% | 0.032 [3] |
| Unchanged sweet foods intake | 117 | 37% | 41 | 42% | 69 | 34% | |
| Increased sweet foods intake | 135 | 42% | 32 | 33% | 93 | 46% | |
| Reduced fluid intake | 21 | 6% | 17 | 17% | | | |
| Unchanged fluid intake | 53 | 16% | 46 | 47% | | N/A | |
| Increased fluid intake | 45 | 14% | 35 | 36% | | | |
| Decreased high-fat foods intake | 46 | 14% | 41 | 42% | | | |
| Unchanged high-fat foods intake | 53 | 16% | 41 | 42% | | N/A | |
| Increased high-fat foods intake | 20 | 6% | 16 | 16% | | | |

Note: Share refers to the share of the respective sub-sample (total sample, fasting women, non-fasting women), excluding missing data (if applicable). p-Values are for tests for differences between fasting vs. non-fasting women.

[1] Women fasting 1 or 2 days (N = 21) are set apart into a separate category (see main text). The sum of the numbers of fasting and non-fasting women therefore does not add up to the total sample size.

[2] χ2 test.

[3] Mann-Whitney U test.

birthweight association only appeared for women who reduced or did not change their intake of high-fat content foods during Ramadan.

For sweet food, fluid intake and sleep, no significant differences between the categories appeared, though the associations with birthweight only reached significance for women eating less sweet foods, drinking less and sleeping less. No moderator effects were found when taking gestational age as the dependent variable (see S3 Fig).

## Robustness and heterogeneity

S1 Table revisits the question how the fasting-birthweight association differs by nutritional intake and sleep behavior, by analyzing this question by trimester. Similar to Fig 5, we only find significant negative associations between fasting in the first trimester and birthweight among offspring to women who did not increase the consumption of high-fat content foods. There are also some indications that fasting in combination with a reduction in the intake of fluids and reduced sleep in the third trimester is associated with a lower birthweight. No significant associations between trimester-specific fasting and gestational length were found. Note that the trimester-specific analyses are less precise, since fewer women fall into each category.

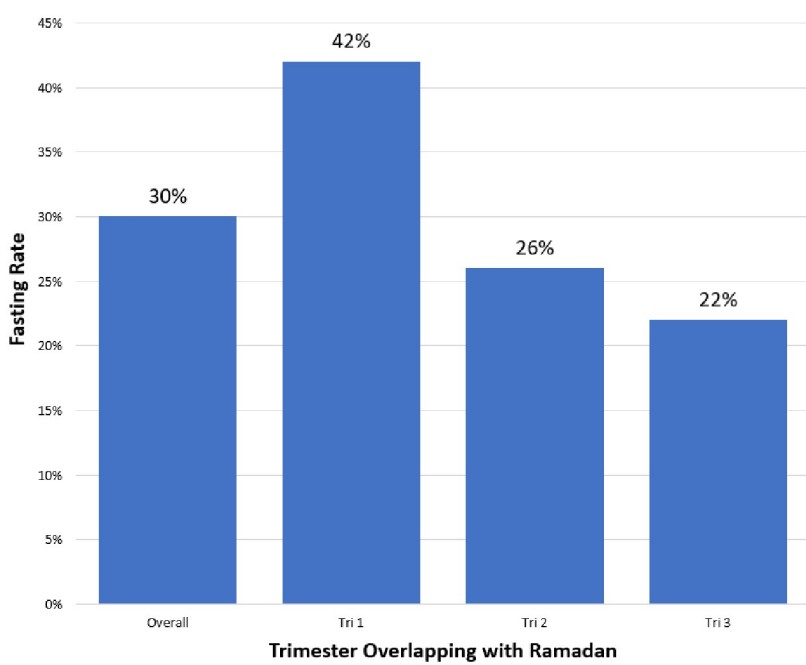

**Fig 2. Fasting rates by trimester.** Overall fasting rate in the sample (left bar) and fasting rates by pregnancy trimester overlap with Ramadan.

The fasting-birthweight association is robust to defining maternal fasting as having fasted at least 1 day during pregnancy (-116·97g, 95% CI: -251·50; 17·57). Results are also robust to reducing the sample to full-term offspring (-181·52g, 95% CI: -316·83; -46·21) and normal-term offspring (-182·85g, 95% CI: -317·72; -47·98).

The results of the Oster test show that it is unlikely that residual confounding has driven the reported associations between fasting and birthweight. Unobserved confounders would not only have to be over eighteen times as important as the included covariates in explaining the outcome, but would have to work in the opposite direction in order to eliminate the detected associations with birthweight ($\delta = -18 \cdot 68$, see S2 File for details).

## Discussion

This study provides first evidence that dietary choices outside of fasting hours may be an important moderator of the associations between Ramadan occurrence during pregnancy and birthweight. Fasting is negatively associated with birthweight, in particular if fasting occurs during the first pregnancy trimester. However, the negative effect of fasting disappears when daily fat intake is simultaneously increased. A possible channel could be that (temporary) caloric deficiencies may be a channel through which intermittent fasting during pregnancy affects offspring birthweight. Since high-fat content foods tend to have higher caloric contents, eating increased amounts of such food might lead pregnant women to reach sufficient daily caloric intakes. Furthermore, since fat has a low glycemic index, increasing consumption of fat on fasting days may help to delay the onset of physical states that are harmful for the fetus (accelerated starvation). Given that Ramadan during pregnancy is a highly sensitive, religious topic and many pregnant Muslims wish to fast for religious reasons [17], this finding is particularly relevant since diets during non-fasting hours are often relatively easily modifiable.

**Table 2. Sample characteristics: Comparison of fasting and non-fasting women using univariate analysis.**

| Category | Total Sample (N = 326) | | Fasting Women (N = 98) | | Non-Fasting Women[1] (N = 207) | | p-value for Diff. |
|---|---|---|---|---|---|---|---|
| | Mean/ Obs.[2] | SD/ Share[3] | Mean/ Obs.[2] | SD/ Share[3] | Mean/ Obs.[2] | SD/ Share[3] | |
| **Birth Outcomes[4]** | | | | | | | |
| Birthweight | 3352 | 531 | 3307 | 511 | 3373 | 540 | 0.312 [5] |
| Low birthweight (<2500g) | 14 | 5% | 4 | 4% | 10 | 5% | 0.818 [6] |
| Gestational age (in weeks) | 39.0 | 1.9 | 39.0 | 2.0 | 39.0 | 1.8 | 0.795 [5] |
| Premature birth (<37 weeks) | 18 | 6% | 6 | 6% | 12 | 6% | 0.852 [6] |
| Low 5-minute APGAR score (<7) | 5 | 2% | 1 | 1% | 4 | 2% | 0.581 [6] |
| Male child | 154 | 51% | 53 | 56% | 101 | 49% | 0.243 [6] |
| **Religiosity** | | | | | | | |
| More religious | 185 | 57% | 78 | 80% | 91 | 44% | <0.001 [6] |
| **Maternal Birth Country** | | | | | | | |
| Germany | 91 | 28% | 11 | 11% | 76 | 37% | <0.001 [6] |
| Syria | 52 | 16% | 27 | 28% | 20 | 10% | |
| Morocco | 48 | 15% | 27 | 28% | 16 | 8% | |
| Turkey | 39 | 12% | 7 | 7% | 31 | 15% | |
| South Asia | 28 | 9% | 8 | 8% | 20 | 10% | |
| Other Arab countries | 23 | 7% | 12 | 12% | 7 | 3% | |
| Somalia | 14 | 4% | 3 | 3% | 11 | 5% | |
| Other | 31 | 10% | 3 | 3% | 26 | 13% | |
| Living in Germany <3 years | 90 | 28% | 40 | 41% | 40 | 19% | <0.001 [6] |
| **Maternal Characteristics** | | | | | | | |
| Age at giving birth | 30.1 | 5.9 | 30.2 | 6.2 | 30.2 | 5.6 | 0.994 [5] |
| Pre-pregnancy BMI | 24.9 | 5.3 | 24.7 | 4.4 | 25.0 | 5.9 | 0.634 [5] |
| Nulliparous | 117 | 36% | 32 | 33% | 75 | 36% | 0.541 [6] |
| Pregnancy risk factors [7] | 45 | 14% | 17 | 17% | 23 | 11% | 0.132 [6] |
| No knowledge of pregnancy during Ramadan | 27 | 8% | 21 | 21% | 4 | 2% | <0.001 [6] |
| Household members fast | 274 | 85% | 95 | 97% | 158 | 77% | <0.001 [6] |
| **Maternal Socio-Economic Status** | | | | | | | |
| Partially/fully employed | 130 | 40% | 26 | 27% | 98 | 47% | 0.001 [6] |
| Technical/university degree | 99 | 30% | 28 | 29% | 65 | 31% | 0.616 [6] |

Note: p-Values are for tests for differences between fasting vs. non-fasting women.

[1] Women fasting 1 or 2 days (N = 21) are set apart into a separate category (see main text). The sum of the numbers of fasting and non-fasting women therefore does not add up to the total sample size.

[2] Means for continuous variables; numbers of observations in category for categorical variables.

[3] Standard deviations for continuous variables; shares of observations in category for categorical variables. Share refers to the share of the respective sub-sample (total sample, fasting women, non-fasting women), excluding missing data (if applicable).

[4] For birth outcomes, the sample is 303 instead of 326 since births to women not consenting to connect medical data, multiple births, and births for which medical data were unretrievable were excluded.

[5] t-test.

[6] $\chi$2 test.

[7] Pregnancy risk factors includes smoking, alcohol consumption, drug use and/or consanguinity.

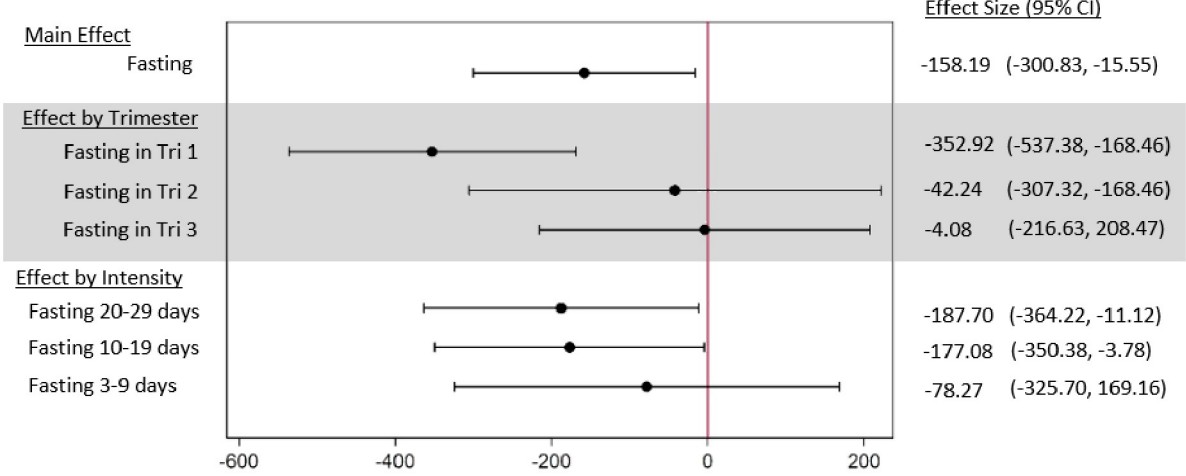

**Fig 3. Fasting and birthweight.** This figure shows the results of three adjusted regressions. The reference group are non-fasting women. Birthweight is measured in grams.

A part of the literature on the health implications of Ramadan during pregnancy mainly finds associations among those for whom Ramadan during pregnancy occurs in early pregnancy [16]. Our finding that fasting rates are highest in the first pregnancy trimester might partly explain these previous findings, as some studies use intent-to-treat designs in which Ramadan exposure is measured as the occurrence of a Ramadan during pregnancy rather than via actual fasting. Beyond that, our analyses also showed indications for the moderating effect of simultaneously increased fat intake to be concentrated among those who fasted during the first pregnancy trimester (S1 Table). This implies that prenatal counselling on the risk of adverse offspring health outcomes in response to Ramadan fasting during pregnancy, and the potential moderating role of dietary intake, should be provided in early pregnancy, or pre-conception.

On the other hand, these results do not allow conclusions about Ramadan fasting during later pregnancy trimesters. A substantial share of women do fast during the second and third

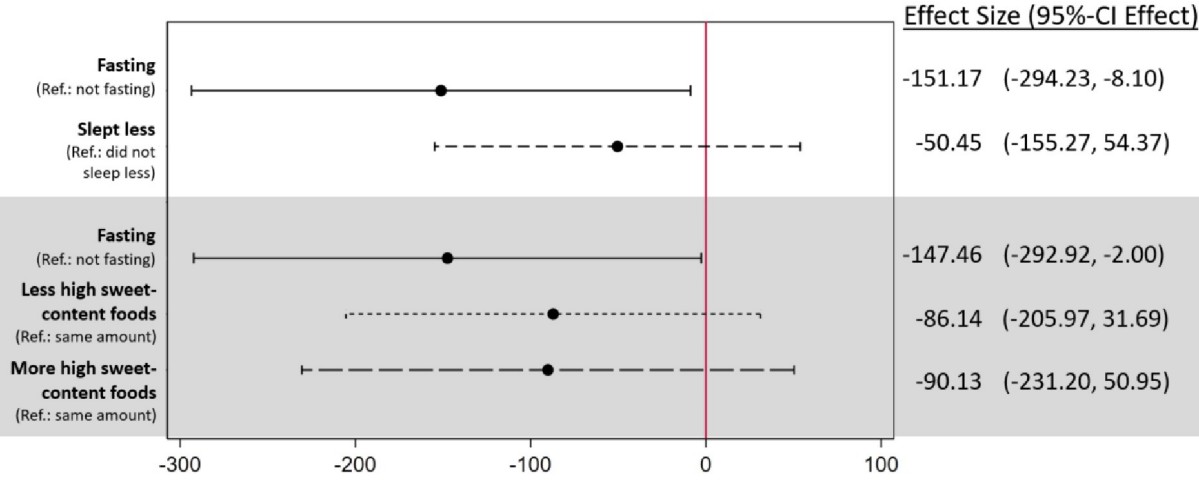

**Fig 4. Fasting, sleep and dietary adaptations and birthweight.** This figure shows the results of two adjusted regressions. The respective reference groups are indicated in the figure. Birthweight is measured in grams.

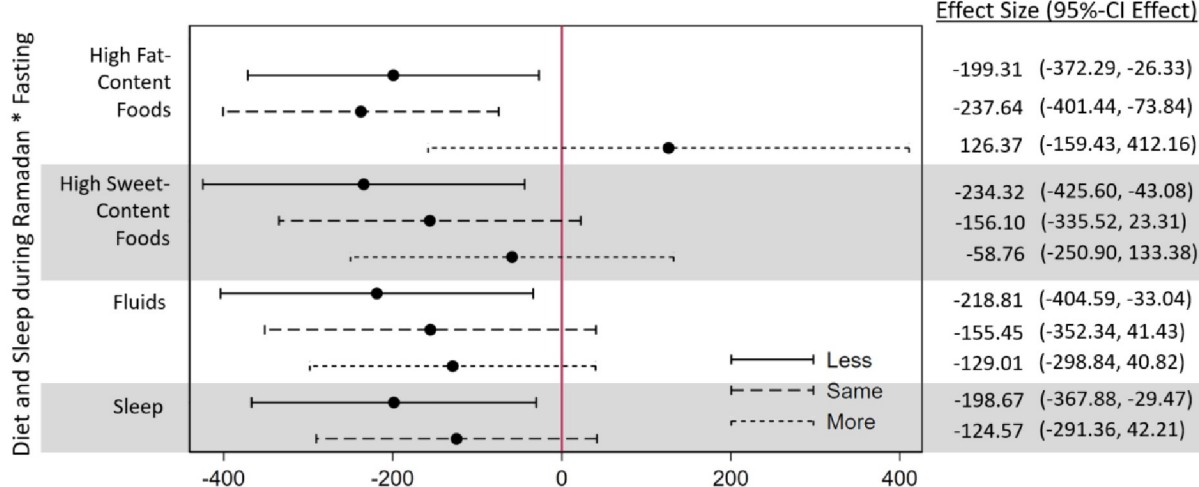

**Fig 5. Regressions of birthweight on fasting interacted with nutritional intake and sleep during Ramadan.** This figure shows the results of four adjusted regressions. In each regression, the fasting variable is interacted with the depicted variable. The reference group is always the offspring of mothers who did not fast. Birthweight is measured in grams.

trimester. The absence of adverse birth outcomes does not preclude that health effects could become manifest at a later point. Previous research found associations between Ramadan during pregnancy in later pregnancy trimesters and various–often chronic–health conditions along the life course [9, 12–25, 30–33]. Moreover, the trimester-specific moderation analyses (S1 Table) suggest that fasting in the third pregnancy trimester could be associated with lower birthweights, if fasting women simultaneously consume less fluids or sleep less than usually. Whether a specific diet during Ramadan also prevents adverse long-run effects, and which pregnancy trimesters matter most, thus remains to be further investigated.

Investigating the potential moderating role of diet and sleep during Ramadan might also shed light on why, in contrast to the literature on long-term health, the evidence on Ramadan during pregnancy and birth outcomes remains less conclusive. A potential explanation why associations with birth outcomes are found in some populations [6, 9, 34] and not in others [19, 35–37] is that the practice of Ramadan differs across and within Muslim communities, leading to context-specific adjustments to Ramadan in terms of fasting behavior, sleep disruptions and stress experiences [7, 37]. Moreover, dietary intake during non-fasting hours differs both between and within countries and ranges from increased dietary diversity during Ramadan to energy deficits among pregnant fasting women [7, 19]. It might be that these factors beyond the binary fasting decision influence whether Ramadan fasting during pregnancy is associated with offspring health a birth.

Previous survey studies were not able to control for covariates such as country of birth, maternal BMI, risky behavior or religiosity [19, 35]. This may also explain some of the inconclusive results from previous research. I.e. the raw mean difference in birthweights between the fasting and the non-fasting group was not significant, and quantitatively considerably smaller than in the controlled regression [36].

## Strengths and limitations

We for the first time provide data on an entire cross-section of Muslim offspring whose time in utero overlapped with a Ramadan (i.e. from conception to birth during Ramadan). Covering

an entire cross-section renders our sample less selective than previous studies, which tend to oversample women in later stages of pregnancy.

Another advantage of our study is the high quality of the data–linking survey with hospital medical data–in conjunction with a systematic and representative sampling design and a high response rate of 72%. In the sample of pregnant Muslims delivering in Mainz, 30% had a higher qualification, and 40% were employed prior to their maternal leave (Table 2). These figures are consistent with the average of people with a Muslim background in Germany [38]. The overall female labor force participation rate, independent of religious background, is at 74.6% and the proportion of women with a vocational degree at 83% in Germany [39].

We approached Muslim mothers in the obstetric wards, where 98% of children in Germany are born [40]. Hospital data allowed us to exactly classify Ramadan exposure by pregnancy trimester based on physical examination. Besides being able to control for a large set of confounders, we also apply the Oster statistical analysis [41] to show that residual confounding is unlikely to have driven our results.

Some limitations should be noted that are mainly due to the fact that we aimed to approach all Muslims in one city who were pregnant in a given year. In order to achieve the goal of approaching all pregnant Muslims in a given year in an entire city, surveys could only be conducted retrospectively. As women in Germany do not visit a hospital at regular intervals during pregnancy, it is impossible to recruit a representative sample of women in all stages of pregnancy who are willing to fill out such diaries during an upcoming Ramadan; especially when the sample should be large enough to conduct multiple linear regression analysis on their linked birth outcomes. This implied that nutritional intake and sleep patterns could only be asked using categorizations rather than via detailed food and sleep diaries. When measuring adjustments to diet and sleep patterns during Ramadan, we had to resort to retrospective measures rather than sleep and food diaries. However, this is more than any previous study has been able to do. From our pilot study we learned that women found it difficult to give estimates of numbers or amounts, but that relative measures (sleeping more or less; consuming more or less, etc.) led to much higher quality responses and less nonresponse. Moreover, we rely on data for one birth cohort and associations potentially differ by the number of hours fasted [9] since in Germany, fasting hours vary with the season into which Ramadan falls. In 2017, Ramadan took place in May and June and thus coincided with long fasting durations in Germany (up to 18 hours).

## Conclusion

This study finds that dietary choices may moderate the associations between intermittent fasting during pregnancy and newborn health. Thereby, additional research is needed to assess the roles of specific (macro)nutrients and food groups, based on which specific recommendations for dietary choices for pregnant Muslims wishing to fast during Ramadan can be developed. This also includes measurements of caloric intakes during Ramadan. Each year millions of Muslim offspring with intrauterine exposure to Ramadan fasting are born. Our study highlights that research on culture-specific habits and traditions is pivotal in order to promote a healthy start to life for all children.

## Supporting information

**S1 File. Questionnaire "Pregnancy during Ramadan".**
(PDF)

**S2 File. Background: Oster method.**
(DOCX)

**S1 Fig. Fasting and gestational age at birth (in weeks).** This figure shows the results of three adjusted regressions. The reference group are non-fasting women. Gestational age at birth is measured in completed weeks of gestation.
(DOCX)

**S2 Fig. Fasting, sleep and dietary adaptations and gestational age at birth in weeks.** This figure shows the results of two adjusted regressions. The respective reference groups are indicated in the figure. Gestational age is measured in completed weeks of gestation.
(DOCX)

**S3 Fig. Effect of fasting on gestational age at birth (in weeks) interacted with dietary intake and sleep during Ramadan.** This figure shows the results of four adjusted regressions. In each regression, the fasting variable is interacted with the depicted variable. The reference group is always the offspring of mothers who did not fast. Gestational age at birth is measured in completed weeks.
(DOCX)

**S1 Table. Regressions of birthweight on fasting interacted with nutritional intake and sleep during Ramadan, by pregnancy trimester of overlap with Ramadan.** Note: Each column shows the results from a separate regression. Each regression includes interaction terms of the fasting by trimester indicators with changes in dietary intake (columns 1–3) and sleep patterns (column 4). Note that for sleep patterns, the comparison is only less sleep vs. unchanged sleep patterns, whereas changes in dietary intake were subdivided into the answer categories less/same/more. Robust standard errors are reported in parentheses. Significance levels: ***p<0.01, **p<0.05, *p<0.1.
(DOCX)

## Acknowledgments

We are indebted to all the patients who participated in this research. We would also like to thank the obstetric wards in Mainz (University Medical Center Mainz, Katholisches Klinikum Mainz), as well as the staff of the Mainz Birth Registry and the University Medical Center Obstetric Archive.

This work could not have been realized without our student assistants who conducted interviews in German, Turkish, Arabic and English: Asmaa Alhamoud, Yara Al-Zamel, Asiye Balci, Ayse Gün, Ranna Salahié, Kheira Sebbane & Hatem Yilmaz.

## Author Contributions

**Conceptualization:** Fabienne Pradella, Birgit Leimer, Anja Fruth, Annette Queißer-Wahrendorf, Reyn Joris van Ewijk.

**Data curation:** Fabienne Pradella, Birgit Leimer, Anja Fruth, Annette Queißer-Wahrendorf.

**Formal analysis:** Fabienne Pradella, Birgit Leimer, Reyn Joris van Ewijk.

**Funding acquisition:** Reyn Joris van Ewijk.

**Investigation:** Fabienne Pradella, Birgit Leimer, Anja Fruth.

**Methodology:** Fabienne Pradella, Birgit Leimer, Reyn Joris van Ewijk.

**Project administration:** Fabienne Pradella, Birgit Leimer, Anja Fruth, Annette Queißer-Wahrendorf.

**Resources:** Fabienne Pradella, Birgit Leimer.

**Supervision:** Reyn Joris van Ewijk.

**Validation:** Fabienne Pradella, Birgit Leimer, Reyn Joris van Ewijk.

**Visualization:** Fabienne Pradella, Birgit Leimer.

**Writing – original draft:** Fabienne Pradella, Birgit Leimer, Reyn Joris van Ewijk.

**Writing – review & editing:** Fabienne Pradella, Birgit Leimer, Anja Fruth, Annette Queißer-Wahrendorf, Reyn Joris van Ewijk.

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
