## [Decision Letter · Decision Letter 0]

28 Nov 2022

PONE-D-22-10907RAMADAN DURING PREGNANCY AND NEONATAL HEALTH – FASTING, DIETARY COMPOSITION AND SLEEP PATTERNSPLOS ONE

Dear Dr. Pradella,

Thank you for submitting your manuscript to PLOS ONE. After careful consideration, we feel that it has merit but does not fully meet PLOS ONE’s publication criteria as it currently stands. Therefore, we invite you to submit a revised version of the manuscript that addresses the points raised during the review process.

We look forward to receiving your revised manuscript.

Kind regards,

Sultana Monira Hussain

Academic Editor

PLOS ONE

Journal Requirements:

Reviewers' comments:

Reviewer's Responses to Questions

**Comments to the Author**

1. Is the manuscript technically sound, and do the data support the conclusions?

Reviewer #1: Yes

Reviewer #2: Yes

2. Has the statistical analysis been performed appropriately and rigorously? 

Reviewer #1: Yes

Reviewer #2: Yes

3. Have the authors made all data underlying the findings in their manuscript fully available?

Reviewer #1: No

Reviewer #2: Yes

4. Is the manuscript presented in an intelligible fashion and written in standard English?

Reviewer #1: Yes

Reviewer #2: Yes

5. Review Comments to the Author

Reviewer #1: I would like to congratulate the authors for conducting very important research on the effects of prenatal Ramadan exposure on child health. Indeed, the current debate is on heterogenous results of the exposure on neonatal indicators. Most studies that did not find evidence of detrimental effects on birth weight and gestational age were either from small sample studies or from large sample studies, but without sufficient covariates.

The large data from Burkina Faso with carefully adjusted models allowed Schoep et al. (2018) to show that in a low-income setting, prenatal Ramadan exposure induced child mortality. Furthermore, using pooled sample data from Indonesia, Kunto & Mandemakers (2019) have shown that weak evidence of negative effects in early childhood develop when the child aged and only become evident once he/she entered late adolescence. However, these two recent studies seemed unable to satisfy discussion on the topic.

The authors’ manuscript adds to the literature by analysing two probable channels, maternal dietary intake and maternal sleeping pattern during Ramadan, that may mask the actual effects of prenatal Ramadan exposure on neonatal indicators. The manuscript was written nicely and anticipated most of my questions on the topic: covariates, robustness, concern on probable omitted variable bias, and explanations on why using categorical variables instead of continuous scale for dietary intakes/sleeping hours. However, I still have very few remarks to improve the manuscript as follow:

1. How long is the daylight hours during Ramadan 2017 in Mainz? It is an important info for the readers could perceived how severe is the accelerated starvation that may drive the effects. Please include this in the text along with socioeconomic context of Muslim community in Mainz.

2. I expect the authors to write explicitly in the manuscript on whether it is safe to do maternal Ramadan fasting in later stages of pregnancy. Considering that:

a. The authors’ finding that maternal fasting rate is significantly higher in the first trimester of pregnancy, and

b. the probability that previous studies did not find effects of prenatal Ramadan exposure in later stages of pregnancy because of maternal fasting rate bias related to the trimester of pregnancy.

3. Did the authors have explored moderating effects of dietary intake and sleeping pattern on the effects of fasting habits to birth weight/gestational age; with and without trimester of pregnancy as one of the independent variables. This step might help the authors to convincingly formulate policy recommendation related to my #2 comment.

Reviewer #2: In this work by Pradella and colleagues, the authors considered the role of dietary composition and sleep patterns during Ramadan that often overlooked in studies exploring the Ramadan during pregnancy effects on health. Overall, the manuscript is written clearly, and the analyses are sound. I am listing below a couple of comments and suggested edits in relation to this work.

Introduction:

- There are two already published systematic reviews on short-term and long-term effects on health which I think worth to be introducing rather than citing each study for different health outcomes. Therefore, I suggest reading and adding these as references (https://doi.org/10.1186/s12884-018-2048-y and https://doi.org/10.3390/nu13124511).

Methods:

- I could not find an explanation for why women participating in the study who had multiple births were excluded from the regression analysis.

- Among 326 included Muslim women in this study, was the data on existing health conditions which can be contributing to birthweight also collected and adjusted for? If not, is there any explanation for not considering this as I guess the hospital medical records included this information?

6. PLOS authors have the option to publish the peer review history of their article (what does this mean?). If published, this will include your full peer review and any attached files.

Reviewer #1: No

Reviewer #2: No

---

## [Author Response · Author response to Decision Letter 0]

28 Dec 2022

Response to Reviewers

We would like to thank the reviewers very much for their constructive comments. We are pleased to hear that you appreciate the quality of our paper and the contribution that we try to make to the literature. Below we reply to each of your comments point-by-point.

Reviewer #1: 

1. How long is the daylight hours during Ramadan 2017 in Mainz? It is an important info for the readers could perceived how severe is the accelerated starvation that may drive the effects. Please include this in the text along with socioeconomic context of Muslim community in Mainz.

Thank you for raising these important issues. In the new version, we specify how long the fasting period was during Ramadan 2017. Below we copy in the new text (line 366 ff. – manuscript version with tracked changes):

"Moreover, we rely on data for one birth cohort and associations potentially differ by the number of hours fasted9 since in Germany, fasting hours vary with the season into which Ramadan falls. In 2017, Ramadan took place in May and June and thus coincided with long fasting durations in Germany (up to 18 hours)."

We also included information on the socioeconomic background (line 341 ff.):

"In the sample of pregnant Muslims delivering in Mainz, 30% had a higher qualification, and 40% were employed prior to their maternal leave (Table 2). These figures are consistent with the average of people with a Muslim background in Germany. The overall female labor force participation rate, independent of religious background, is at 74.6% and the proportion of women with a vocational degree at 83% in Germany."

2. I expect the authors to write explicitly in the manuscript on whether it is safe to do maternal Ramadan fasting in later stages of pregnancy. Considering that:

a. The authors’ finding that maternal fasting rate is significantly higher in the first trimester of pregnancy, and

b. the probability that previous studies did not find effects of prenatal Ramadan exposure in later stages of pregnancy because of maternal fasting rate bias related to the trimester of pregnancy.

Thank you for this comment. In response to this valid point, we included a discussion on which conclusions regarding the safety of Ramadan fasting and related policy recommendations we can infer from this study. In this new text, we refer to a new Appendix Table VI which we added in response to your comment number 3 (see below). The new text, which is placed in the Discussion section, reads:

"A part of the literature on the health implications of Ramadan during pregnancy mainly finds associations among those for whom Ramadan during pregnancy occurs in early pregnancy. Our finding that fasting rates are highest in the first pregnancy trimester might partly explain these previous findings, as some studies use intent-to-treat designs in which Ramadan exposure is measured as the occurrence of a Ramadan during pregnancy rather than via actual fasting. Beyond that, our analyses also showed indications for the moderating effect of simultaneously increased fat intake to be concentrated among those who fasted during the first pregnancy trimester (Online Appendix VI). This implies that prenatal counselling on the risk of adverse offspring health outcomes in response to Ramadan fasting during pregnancy, and the potential moderating role of dietary intake, should be provided in early pregnancy, or pre-conception.

On the other hand, these results do not allow conclusions about Ramadan fasting during later pregnancy trimesters. A substantial share of women do fast during the second and third trimester. The absence of adverse birth outcomes does not preclude that health effects could become manifest at a later point. Previous research found associations between Ramadan during pregnancy in later pregnancy trimesters and various – often chronic – health conditions along the life course. Moreover, the trimester-specific moderation analyses (Online Appendix VI) suggest that fasting in the third pregnancy trimester could be associated with lower birthweights, if fasting women simultaneously consume less fluids or sleep less than usually. Whether a specific diet during Ramadan also prevents adverse long-run effects, and which pregnancy trimesters matter most, thus remains to be further investigated."

3. Did the authors have explored moderating effects of dietary intake and sleeping pattern on the effects of fasting habits to birth weight/gestational age; with and without trimester of pregnancy as one of the independent variables. This step might help the authors to convincingly formulate policy recommendation related to my #2 comment.

Thank you for bringing up this interesting point. We had in fact already run exactly these analyses that you suggest. However, we had to make a choice with respect to the results to include in the paper and had originally decided not to include them. Since we agree to the relevance with respect to the ongoing discussions on trimester-specific effects, we have now added them to the paper as an appendix table (Online Appendix VI). 

We agree that this table allows the reader to get a better understanding of the background dynamics of our results and hope that the inclusion of these results makes our paper stronger. Since this is the first study to include dietary intake and sleep patterns in analyses on the associations between Ramadan fasting during pregnancy and offspring health outcomes, we believe that showing this analysis in the appendix might also be important for comparison with future studies.

We added the following texts about these analyses to our manuscript:

Line 177 ff. (Methods section)

"In an additional analysis, we interact trimester-specific exposure with sleep and dietary intake patterns in order to investigate if potential associations are concentrated in specific pregnancy phases."

Line 263 ff. (Results section)

"Online Appendix VI revisits the question how the fasting-birthweight association differs by nutritional intake and sleep behavior, by analyzing this question by trimester. Similar to Figure 5, we only find significant negative associations between fasting in the first trimester and birthweight among offspring to women who did not increase the consumption of high-fat content foods. There are also some indications that fasting in combination with a reduction in the intake of fluids and reduced sleep in the third trimester is associated with a lower birthweight. No significant associations between trimester-specific fasting and gestational length were found. Note that the trimester-specific analyses are less precise, since fewer women fall into each category."

Reviewer #2: In this work by Pradella and colleagues, the authors considered the role of dietary composition and sleep patterns during Ramadan that often overlooked in studies exploring the Ramadan during pregnancy effects on health. Overall, the manuscript is written clearly, and the analyses are sound. I am listing below a couple of comments and suggested edits in relation to this work.

Introduction:

- There are two already published systematic reviews on short-term and long-term effects on health which I think worth to be introducing rather than citing each study for different health outcomes. Therefore, I suggest reading and adding these as references (https://doi.org/10.1186/s12884-018-2048-y and https://doi.org/10.3390/nu13124511).

Thank you for pointing to these two recent systematic reviews. We read them carefully and now refer to the two reviews in the introduction.

Methods:

- I could not find an explanation for why women participating in the study who had multiple births were excluded from the regression analysis.

Thank you for this comment, we agree that this is relevant background information for the reader. We included an explanation in the methods section (line 87 ff.– manuscript version with tracked changes):

"All Muslims who delivered their singleton newborn in one of the two obstetric wards in Mainz and whose pregnancy overlapped with Ramadan 2017 were eligible for participation. Muslim women without overlap with Ramadan 2017 were excluded from the sample, as well as women who preregistered for delivery but did not deliver their baby in Mainz (Figure 1). Multiple births were excluded from the regression analysis since multiple births form a special group in terms of prenatal care provision as well as neonatal health outcomes, including low birth weight and lower gestational age."

- Among 326 included Muslim women in this study, was the data on existing health conditions which can be contributing to birthweight also collected and adjusted for? If not, is there any explanation for not considering this as I guess the hospital medical records included this information?

Thank you for pointing out this important issue. Maternal health conditions can either be pre-existing ones, or those arising during pregnancy. Unfortunately, hospital data confidentiality regulations did not allow us to link our data to conditions that already existed before the pregnancy. (This information is stored separately from birth documentation.) To add questions about pre-existing conditions to our survey would have made the survey too long (as many potential health conditions may exist). We worried that this would lead women to refuse participation in our survey.

Maternal health conditions that arise during pregnancy might themselves be a result by the mother’s fasting. (For example, it could be imagined that pregnancy diabetes might sometimes result from fasting.) In that sense, they are dependent rather than independent variables and adjusting for such variables would bias our effects of interest.

It would of course have been interesting to study whether pregnancy health conditions such as gestational diabetes are indeed induced by Ramadan fasting. However, a very large share of women had missing values on the relevant variables in the hospital data, so that such analyses would have remained underpowered. We have to leave this interesting research question for future research.

---

## [Decision Letter · Decision Letter 1]

16 Jan 2023

RAMADAN DURING PREGNANCY AND NEONATAL HEALTH – FASTING, DIETARY COMPOSITION AND SLEEP PATTERNS

PONE-D-22-10907R1

Dear Dr. Pradella,

We’re pleased to inform you that your manuscript has been judged scientifically suitable for publication and will be formally accepted for publication once it meets all outstanding technical requirements.

Kind regards,

Sultana Monira Hussain

Academic Editor

PLOS ONE

Additional Editor Comments (optional):

Reviewers' comments:

Reviewer's Responses to Questions

**Comments to the Author**

1. If the authors have adequately addressed your comments raised in a previous round of review and you feel that this manuscript is now acceptable for publication, you may indicate that here to bypass the “Comments to the Author” section, enter your conflict of interest statement in the “Confidential to Editor” section, and submit your "Accept" recommendation.

Reviewer #1: All comments have been addressed

Reviewer #2: All comments have been addressed

2. Is the manuscript technically sound, and do the data support the conclusions?

Reviewer #1: Yes

Reviewer #2: Yes

3. Has the statistical analysis been performed appropriately and rigorously? 

Reviewer #1: Yes

Reviewer #2: Yes

4. Have the authors made all data underlying the findings in their manuscript fully available?

Reviewer #1: Yes

Reviewer #2: Yes

5. Is the manuscript presented in an intelligible fashion and written in standard English?

Reviewer #1: Yes

Reviewer #2: Yes

6. Review Comments to the Author

Reviewer #1: The revisions has made the manuscript more refined. The appendix VI is very interesting as it provides important evidence to what others have been speculate on the mechanisms that link fasting and the offspring's health/growth indicators (e.g. Ewijk 2011, 2014; Kunto & Mandemakers, 2019). This manuscript is certainly worth a publication.

Reviewer #2: Thank you for the revised version. The main contribution of this paper is the fact that the authors considered the role of dietary composition and sleep patterns during Ramadan which is often overlooked in studies exploring the Ramadan during pregnancy effects on health. Overall, the manuscript is written clearly, and the analyses are sound.

Further, the authors have addressed all my comments and I recommend this paper be accepted.

7. PLOS authors have the option to publish the peer review history of their article (what does this mean?). If published, this will include your full peer review and any attached files.

Reviewer #1: **Yes: **Yohanes Sondang Kunto

Reviewer #2: No
